# Bio-Polyester/Rubber Compounds: Fabrication, Characterization, and Biodegradation

**DOI:** 10.3390/polym15122593

**Published:** 2023-06-07

**Authors:** Carina Frank, Anita Emmerstorfer-Augustin, Thomas Rath, Gregor Trimmel, Manfred Nachtnebel, Franz Stelzer

**Affiliations:** 1Acib-GmbH, Krenngasse 32, A-8010 Graz, Austria; 2Institute for Chemistry and Technology of Materials, Graz University of Technology, Stremayrgasse 9, A-8010 Graz, Austria; thomas.rath@tugraz.at (T.R.); gregor.trimmel@tugraz.at (G.T.); 3Institute for Molecular Biotechnology, Graz University of Technology, NAWI Graz, BioTechMed-Graz, Petersgasse 14, A-8010 Graz, Austria; 4Graz Centre for Electron Microscopy, Steyrergasse 17, A-8010 Graz, Austria; manfred.nachtnebel@felmi-zfe.at

**Keywords:** bio-polyester, poly(hydroxyalkanoates), poly(3-hydroxybutyrate-*co*-3-hydroxyvalerate), natural rubber, compounding, biodegradation

## Abstract

Biobased and biodegradable polymers (BBDs) such as poly(3-hydroxy-butyrate), PHB, and poly(3-hydroxybutyrate-*co*-3-hydroxyvalerate) (PHBV) are considered attractive alternatives to fossil-based plastic materials since they are more environmentally friendly. One major problem with these compounds is their high crystallinity and brittleness. In order to generate softer materials without using fossil-based plasticizers, the suitability of natural rubber (NR) as an impact modifier was investigated in PHBV blends. Mixtures with varying proportions of NR and PHBV were generated, and samples were prepared by mechanical mixing (roll mixer and/or internal mixer) and cured by radical C–C crosslinking. The obtained specimens were investigated with respect to their chemical and physical characteristics, applying a variety of different methods such as size exclusion chromatography, Fourier-transform infrared spectroscopy (FTIR), scanning electron microscopy (SEM), thermal analysis, XRD, and mechanical testing. Our results clearly indicate that NR–PHBV blends exhibit excellent material characteristics including high elasticity and durability. Additionally, biodegradability was tested by applying heterologously produced and purified depolymerases. pH shift assays and morphology analyses of the surface of depolymerase-treated NR–PHBV through electron scanning microscopy confirmed the enzymatic degradation of PHBV. Altogether, we prove that NR is highly suitable to substitute fossil-based plasticizers; NR–PHBV blends are biodegradable and, hence, should be considered as interesting materials for a great number of applications.

## 1. Introduction

While plastics have become highly valued for their long-lasting versatile and functional use, cheap price, and good commercial availability, their very slow degradation rate has challenged society with the problematic accumulation of plastic waste leading to big environmental concerns. Carelessly disposed plastic waste and “microplastics” are the most concerning part of environmental pollution. In 2020, 367 million tons of synthetic polymers were produced worldwide [1]. Furthermore, 40% of all plastic products are designed as short-term or even single-use products that contribute to a constantly growing mountain of plastic waste [2]. Of all plastics discarded since 1952, 91% have never been recycled, ending up in environmental ecosystems or being dumped in landfills. Currently, only 14% of all plastic is recycled, whereas only 2% is optimally recycled; the remaining 12% is downcycled [3]. This means that the remaining 86% of consumer plastic takes up substantial volume because of its low density, can easily be blown everywhere by winds, and is lost from society as a resource. Using fossil resources for producing single-use products is legitimately considered pure wastage, especially when considering the limited availability, increasing pricing, and contribution to climate change.

Therefore, the urgency for alternative end-of-life scenarios is increasing, and a growing interest in sustainable products can be observed. Biobased and biodegradable polymeric materials (“biodegradable biobased plastics”, BBDs) are considered an appropriate means to realize [4] the wish for environmentally friendly, sustainable plastics. Currently, bioplastics account for less than 1% of the total world plastic generated. However, the market for bioplastics is continuously growing. The European Bioplastics Association predicts that the global bioplastic production capacities will increase from around 2.11 million tons in 2020 to approximately 2.87 million tons in 2025 [5]. Actually, new materials—especially commodity plastics—can be easily reused or recycled. However, when unintendedly discarded, it would be even better for them to “disappear” in the environment through degradation. When used for food packaging, the plastic film should be as easily compostable as the biowaste itself. Consequently, the use of such biopolymers (biologically produced and biodegradable polymers) is increasing, but their properties often have to be modified drastically to meet market demands.

The research presented in this paper gives a broad overview of new compound developments based on biologically produced polyesters together with natural rubber. Such compounds have gained increasing interest recently as blends of a crystalline bio-thermoplastic and a natural elastomer because of their sustainable resources and biodegradability. We present a set of encouraging results which, nevertheless, have to be considered preliminary; however, they will be the topic of further detailed investigations with respect to sustainable applications, e.g., degradability of the rubber partition in these blends.

Polyhydroxyalkanoates (PHAs) represent a typical class of biomaterials for the future due to their physicochemical properties, biodegradability, and biocompatibility [6,7]. PHAs are produced through the bacterial fermentation of sugars, lipids, organic waste, or even CO_2_ [8,9,10]. One of the best-described PHAs is poly(3-hydroxy butyrate (PHB), which is a linear polyester with similar properties to polypropylene (PP), such as a high elasticity modulus and high crystallinity. However, PHB is very brittle and has a very poor thermal stability. The thermal decomposition starts at temperatures close to the melting point, which limits its processability [11]. One strategy to improve this is the incorporation of 3-hydroxyvalerate (3HV) units into the PHB molecule, resulting in a copolymer, poly(3-hydroxybutyrate-*co*-(3-hydroxyvalerate), widely known as PHBV. However, the physical and mechanical properties of PHBV greatly depend on the 3HV content. Thus, the melting point of PHBV decreases as the fraction of 3HV in the copolymer increases, which broadens the processing window. Additionally, the crystallinity decreases, resulting in a more flexible structure, which generally improves the mechanical properties of the material [11,12]. However, the controlled production of a well-defined copolymer is quite tricky in the biotechnological production process and increases the price remarkably [12]. Furthermore, modification of polymer properties via the biotechnological production process is expensive and time-consuming. Therefore, blending the copolymer with polymeric modifiers such as poly(butylene adipate-co-terephthalate) is common practice. To substitute this mainly fossil-based additive, other natural materials (e.g., plasticizing oils, other biobased components or polymers such as PLA [13], or natural rubber NR) could be a more sustainable strategy to improve the properties and the processability of PHB/PHBV. NR is an elastomer with high elasticity and yield strength, and it provides an unmatched combination of tensile strength, elasticity, memory, and softness [14]. Recent studies of PHB/PHBV with natural rubber have shown significant improvements in their properties [15,16]. Therefore, the incorporation of NR into PHBV is considered a promising route to improve the flexibility and toughness of PHBV [15]. As NR is biobased and biodegradable, the “green” nature of the blends is maintained.

In addition, crosslinking allows the blend to reach high strains under stress through the formation of strong fibrils and molecular networks [15]. For rubber compounds, this crosslinking is known as “vulcanization”, a chemical process initiated by either sulfur or free radicals, e.g., from peroxides. In cases of polymers with alkyl side-chains, this crosslinking might generate covalent bonds between different phases, e.g., the rubber segments and the side-chains of the thermoplastic polymer. Hence, even in phase-separated blends, a recrystallization of the thermoplastic component can be avoided or remarkably reduced.

It was already mentioned above that new polymeric materials should be both biobased and biodegradable; however, in spite of degradability, they should also be reusable or even recyclable. This assumes a deep knowledge of the physical/thermal stability on one hand and of degradation on the other hand. Therefore, very detailed characterization of the degradability of new polymeric materials is important. The degradability of PHAs is generally accepted and quite well investigated and understood [17,18,19]. However, natural rubber—despite being a natural polymer—does not biodegrade as easily, especially in its vulcanized form. Only limited insights into the enzymatic degradation of latex and latex products have been gained up to now [20,21,22]. As “micro-“ or (more correctly) “nano-rubber” is known to be the dominant partition of environmental fine particulate matter, caused by traffic via the abrasion of tires [23], the investigation of the possibility of biodegradation of rubber in the environment is both a great challenge and an environmental demand.

Thus, we decided to investigate the biodegradability of rubber compounds in parallel with the development of PHBV/NR compounds. Because of the often extremely time-consuming experiments with regard to composting or degradation of biomaterials in the environment, the development of faster in vitro methods seemed reasonable, and we are happy to present some preliminary results of new in vitro methods in this paper.

## 2. Materials and Methods

### 2.1. Materials

Standard natural rubber (NR) was provided by Semperit Technische Produkte GmbH (Vienna, Austria). Two different types of poly(3-hydroxybutyrate-co-3-hydroxyvalerate) (PHBV) (PHi001 with approximately 5 wt.% hydroxyvalerate (HV) units and a comparably high amount (>40 wt.%) of PBAT (poly(butylene adipate-*co*-terephthalate)) as an impact modifier, and PHi002 with approximately 5 wt.% hydroxyvalerate units, without PBAT) were provided by NaturePlast (Ifs, France). PBAT is an often used biodegradable but fossil-based thermoplastic modifier. The average molecular weight (M_w_) of PHi001 was found to be 61.9 × 10^3^ (PDI: 1.99) determined by gel permeation chromatography (GPC). PHi002 has an average weight of 65.1 × 10^3^ and a PDI of 1.06 [24]. Di(tert-butylperoxyisopropyl) benzene (Perkadox 14-40B-pd) was purchased from Sigma-Aldrich (Vienna, Austria). Both rubber and Perkadox were used as received. The PHBV pellets were dried for 4 h at 60 °C to remove moisture.

### 2.2. Blend Preparation

The raw material from NaturePlast was blended with natural rubber in an internal mixer, which was controlled steadily at a temperature of 160 °C for 5 min at 70 rpm. Various ratios of 20:80, 50:50, and 80:20 (wt./wt.) of NR/PHBV were investigated. The corresponding samples were denoted as 20, 50, and 80, respectively. Different amounts of peroxide (Perkadox 14-40B-pd) were added as the curing agent using a two-roll mill at room temperature. Crosslinked plates of the materials were produced using a Collin platen press P 200 PV. Preheated (160 °C) iron molds were filled with samples. Plates were compression-molded at 160 °C for 15 min with a pressure of 40 bar. After the pressing process, the form was cooled down inside the platen press. This process yielded plates of about 1 mm thickness. Plates were granted 24 h relaxation time prior to any further steps. In order to get an overview of the properties for the full range of compositions, we chose PHBV/NR mass ratios of 80/20, 60/40, 40/60, and 20/80 along with the pure polymers for comparison (Table 1). In addition, we investigated the blend with changing continuous phase from PHBV to NR.

### 2.3. Fourier-Transform Infrared (FTIR) Spectroscopy

Samples (NR, PHBV, and NR/PHBV blends) were characterized by a Bruker Alpha-P FT-IR spectrometer. The samples were scanned in attenuated total reflection (ATR) mode ranging from 4000 to 400 cm^−1^ with a resolution of 4 cm^−1^ and 48 scans.

### 2.4. Differential Scanning Calorimetry (DSC)

DSC measurements were accomplished with a DSC 8500 from Perkin Elmer. Analyses were performed under N_2_ atmosphere with a flow rate of 20 mL·min^−1^. About 5 mg of the sample were heated in an aluminum pan from −20 °C to 200 °C (first heating cycle), then cooled again to −20 °C, and afterward heated once more to 200 °C (second heating cycle). The heating/cooling rate was set to 20 and 40 °C·min^−1^, respectively. Glass transition temperatures (*T_g_*), crystallization temperatures (*T_c_*), melting points (*T_m_*), and the associated enthalpies with crystallization and melting (Δ*H_c_* and Δ*H_m_*, respectively) were calculated from the second heating ramp. The percentage crystallinity was calculated from the cooling step after the second heating, using the following equation [25]:(1)$$X\;\left[\%\right]=\frac{\Delta H_{m}}{\Delta H_{m}^{0}}\times 100,$$
where $\Delta H_{m}$ is the enthalpy of fusion in J/g of the experimental peak for each sample, and $\Delta H_{m}^{0}$ is the corresponding melting enthalpy of pure crystalline PHBV (146 J/g) [26].

### 2.5. Thermogravimetric Analysis (TGA)

TGA was performed on a Netzsch STA 449 C. About 4–5 mg of the sample were weighed into an aluminum pan and heated from 20 °C to 550 °C with a heating rate of 10 K·min^−1^. Thermal degradation studies were performed under helium atmosphere with a flow rate of 50 mL·min^−1^.

### 2.6. Rheological Characterization

Curing behaviors of the NR/PHBV blends were analyzed using a Prescott Instrument Ltd. Rheoline Multifunction Rheometer. Rheological characterizations were performed at 160 °C with an amplitude of 0.5° and a frequency of 1.67 Hz for 15 min. Different amounts of peroxide were added to the NR/PHBV blend to characterize the crosslinking process. For the measurement, 3.5 g of material were placed in the rheometer.

### 2.7. Mechanical Properties

Tensile testing was conducted using a Shimadzu universal testing machine AGS-X, operating at a crosshead speed of 100 mm·min^−1^. Crosslinked plates were cut into mechanical testing specimens with dimensions of 75 × 4 × 1 mm (ISO 527), using a hydraulic press and a specimen cutter. The mechanical characteristics of the blends are determined by the parameters of tensile strength and relative elongation at break. The reported standard deviation values were calculated from at least four samples.

### 2.8. X-ray Diffraction

X-ray diffraction patterns were measured on a 600 W Rigaku Miniflex diffractometer using CuK_α_ radiation. The characterizations were performed at room temperature on a zero-background sample holder. The diffraction patterns were recorded in a 2*θ* range of 4–30° with a step width of 0.01° at scan rate of 2°/min.

### 2.9. Determination of the Degree of Crosslinking

The degree of crosslinking was determined by swelling tests. Crosslinked samples of about 0.3 g were put into chloroform for 24 h at room temperature. The samples were removed from the chloroform, patted dry with filter paper, and weighed after 1, 3, 5, and 24 h. The degree of swelling q was calculated via the following equation [27]:(2)$$q=\frac{m_{swollen}-m_{dry}}{m_{dry}}\times 100.$$

### 2.10. In Vitro Cytotoxicity Test

To ascertain the harmonious effect and the potential application of the different crosslinked blends, cytotoxicity levels were investigated. According to ISO 10993-5, a direct contact cytotoxicity assay was performed using MRC-5 cells to evaluate the dehydrogenase activity of the cells exposed to various samples.

To obtain sub-confluent cultures, MRC-5 cells were seeded and cultured 24 h prior to sample exposure. Samples were extracted in cell culture medium for 24 h at 37 °C. Seeding for the 24 h exposures was 16,000 cells/well. Eluates were applied in the following concentrations: pure–1 + 1–1 + 4–1 + 9–1 + 19. Cells were cultured for 24 h (positive control: addition of 1% Triton-X (dissolved 1:1 in 70% ethanol) for 10 min; negative control: cell culture medium). Dehydrogenase activity of the cells was used as an indication for cell viability calculated according to the following formula [28]:(3)$$Dehydrogenase\;Activity\;\left[\%\right]=100\times \frac{\left(A_{490\;nm}\;sample\right)-\left(A_{490\;nm}\;blank\right)}{\left(A_{490\;nm}\;control\right)-\left(A_{490\;nm}\;blank\right)}.$$

### 2.11. Enzymatic In Vitro Degradation

The PHB depolymerase from *Pseudomonas lemoignei* (*Pl*DP) was used for enzymatic degradation of all polymer blends. The gene encoding *Pl*DP was codon-optimized, cloned, and expressed as a C-terminally His_6_-tagged fusion in *E. coli*. Recombinant *Pl*DP was purified by metal affinity chromatography using an ÄKTA pure protein purification system. The activity of PHB depolymerase was determined using a pH indicator-based method through indirect measurement of free hydroxybutyric acid released by hydrolysis of PHB (PHBV). Briefly, 100 mg of material was overlaid with 2 mL of 10 mM KPi buffer (pH 7.4) containing 100 µM fluorescein. Then, 0.1 mg/mL of aqueous *Pl*DP was added to start the reaction. Samples were incubated at 30 °C and 300 rpm for 168 h. The samples were measured at wavelengths of 485 nm and 514 nm. The gain was adjusted to autogain according to the sample with the most fluorescence in order to get about 90% of the maximum signal before detector saturation.

### 2.12. Scanning Electron Microscopy (SEM)

The morphology of the various blends before and after enzymatic degradation was observed by scanning electron microscopy (SEM). For SEM, the samples were directly investigated after the blend preparation. As all samples were nonelectrically conductive, classical high-vacuum SEM was not applicable without adding a conductive layer. To overcome this issue, an environmental SEM (ESEM) was used (FEI ESEM Quanta 450 FEG, Hillsboro-US). This microscope enables the direct investigation of the polymers using so-called “low-vacuum mode” [29]. To prevent an alteration of the polymer due to the interaction with the electron beam [30], a low acceleration voltage of 7.0 kV and moderate beam current were used.

SEM analyses were performed on NP1 50 (PHi001/NR 50:50) and NP2 50 (PHi002/NR 50:50) treated with 0.1 mg/mL of *Pl*DP. As control samples, untreated NP1 50 and NP2 50 were studied. The change in the surface morphology of crosslinked PHBV/NR blends was analyzed after 180 days of enzymatic degradation. Images of each sample were taken at 50×–8000× magnification.

## 3. Results and Discussion

### 3.1. Curing Behavior

When the peroxide is added to the blend and processed at high temperature, a thermal degradation of PHBV occurs simultaneously to crosslinking. Therefore, the optimum crosslinking time must be determined to maximize crosslinking and minimize degradation. Hence, in order to retain crosslinked blends, the processing should be ended before the degradation rate exceeds the crosslinking. This means that a portion of the peroxide remains undecomposed. The cure characteristics at 160 °C of various NR/PHBV blends were tested with various amounts of radical initiator (Figure 1). The maximum torque of the blends increased with the amount of added crosslinker. In the vulcanization process of various blends, the torque S’ increased over time; therefore, the peroxide might not yet have been consumed completely.

By increasing the amount of peroxide from 0% to 6% (Figure 1a,b), the crosslink density of the various blends increased, along with the torque values. This was more evident for NP1 blends, where an increase in peroxide also led to a clear increase in torque from 1.1 to 7.3 dN·m. As expected, blend NP2 50 6% had the highest maximum torque with the value of 39.9 dN·m. Maximum torque can be expressed as a measure of stiffness of rubber; hence, blend NP2 50 6% had the highest stiffness compared to other blends. However, the differences between torque values of NP2 50 4% and 6% were not that large; thus, we used lower amounts (4%) of crosslinker for further testing, as it was expected that higher concentrations might influence the cytotoxicity of the compounds.

In general, it could be observed that the blends with PHi002 (Figure 1b,d) had higher torque values compared to blends with PHi001 (Figure 1a,c). This could be attributed to the fact that PHi002 was lacking the modifying additive and was, therefore, a more rigid material with higher crystallinity and brittleness. In conclusion, peroxide-initiated crosslinking was identified as a good curing system, creating an efficiently vulcanized blend well suited for further studies in several tests.

### 3.2. Crosslinking Density

A preliminary swelling study is indicative of the crosslinking density of different NR/PHBV blends. In this study, swelling experiments were carried out in chloroform for NR/PHBV 50:50 blends crosslinked with different amounts of peroxide at room temperature. The degree of swelling was calculated according to Equation (2). For samples analyzed in this study, swelling increased with time, first rapidly and then slowly, reaching a plateau (Figure 2). All the blend formulations showed swelling ratios of more than 250% within the first 2 h; the uncured sample with 0% crosslinker remained completely soluble. As expected, the largest solvent intake (swelling) was seen for the samples with the lowest peroxide content (2%), while the blends with 6% crosslinker showed the lowest percentage of swelling. It was observed that blends of PHi001 generally achieved higher swelling values. Additionally, the extracted part was poured into a glass petri dish, and the solvent was evaporated. For the remaining films, IR spectra were determined (Figure 3) to observe which parts of the blend are mainly extracted by the solvent. The extractable part of the crosslinked compounds was always far less than 1% (measured by weight). This suggests that crosslinking occurred not only in the rubber phase but also in the PHB phase or at least at the interface between both phases as the PHB phase was no longer extractable after crosslinking.

### 3.3. Fourier-Transformed Infrared (FTIR) Spectroscopy

The FTIR analysis was performed to identify the physical and chemical interactions between the different polymers of the blends. The FTIR spectra for NR, PHBV, and typical NR/PHBV blends with a 50:50 ratio are shown in Figure 3. For neat NR, absorption bands at 2950 cm^−1^, 2922 cm^−1^, and 2844 cm^−1^ were observed. The first peak corresponds to the CH_3_ asymmetric vibrations, while the other two depict asymmetric and symmetric CH_2_, as well as CH_3_ vibrations. Additionally, the small peak, which appeared at 1650 cm^−1^, is due to C=C stretching, while another two sharp peaks at 1448 cm^−1^ and 1370 cm^−1^ correspond to the CH_2_ and CH_3_ bending of NR. On the other hand, the significant peak at 838 cm^−1^ represents =CH out-of-plane bending.

In contrast, the main peak of PHBV (PHi002) occurred at 1720 cm^−1^, which is attributed to symmetric C=O stretching in aliphatic esters. Moreover, C–O–C stretching vibration appeared in the region 1100–1300 cm^−1^. The peaks at 2956 cm^−1^ and 2932 cm^−1^ correspond to the asymmetric and symmetric –CH stretching vibration of the crystalline phase of PHBV [31,32]. Meanwhile, peaks that occurred at 1457 cm^−1^, 1372 cm^−1^, and 826 cm^−1^ belong to the various CH modes, such as asymmetric deformation, symmetric deformation, and bending. As PHi001 is a combination of PHBV and PBAT, peaks which revealed the symmetric stretching vibration of C–O at 1268 cm^−1^ and the bending vibration absorption of the CH plane of the benzene ring at 729 cm^−1^ of PBAT were found.

The comparison of the spectra of NR/PHBV blends with those of pure PHBV and NR confirms that peaks for both NR and PHBV were present in the blends. As an example, the main peaks attributed to PHBV were observed at 1720 cm^−1^, 1457 cm^−1^, 1372 cm^−1^, 2956 cm^−1^, and 2932 cm^−1^. Meanwhile, the peaks for NR were seen at 2844 cm^−1^ and 838 cm^−1^.

The IR spectra of the extracts corresponded to the typical spectra of the PHi001/PHi002 spectra, respectively. This led to the assumption that only the PHBV part of the blend was extracted during the swelling experiment, whereas NR was obviously fully crosslinked and, therefore, insoluble. However, as less than 1% (measured by weight) could be extracted from the material, it can be assumed that the major PHA fraction was also crosslinked.

### 3.4. Mechanical Properties

One of the main purposes of blending PHBV with natural rubber is to increase its flexibility; hence, the mechanical properties of the blends were measured (Figure 4). The addition of natural rubber to the very brittle material PHBV (especially PHi002) led to a very substantial improvement in mechanical properties. The E-modulus and tensile strength were decreased upon adding an elastomer, but elongation at break increased. This was mainly due to the rubber elasticity of NR phase [33], while the crystalline PHBV phase acted as a filler. Similar results were also observed in [15].

The results showed that the brittleness of PHBV could be improved already by low NR loading, leading to a plasticizing effect. The blend PHi001 showed distinctly lower values in E-modulus and tensile strength because of the basic content of PBAT as a modifier. Generally, blends with low NR content (20%) were characterized by a high drop of E-modulus from the high PHBV values to 146.2 and 604.8 MPa, as well as by a high tensile strength of 13 and 14.2 MPa for NP1 and NP2 respectively. The stiffness of the blends was further confirmed by the lowest elongation at break of 76.3% and 6.4%. Blends with PHi002 tended to show higher stiffness. The brittle behavior of PHi002 was attributed to the very high degree of crystallinity of 72% (Table 2) as a consequence of the absence of a modifier.

It is important to emphasize that PHi001 is composed of PHBV and PBAT, which exhibits low elastic modulus and stiffness, but high flexibility and toughness. This makes it ideal for blending with another biodegradable polymer such as PHB of PLA. The introduction of PBAT into blends provoked a significant change in the mechanical properties, resulting in a quite low E-modulus. By adding 20% NR to the blends, the E-modulus value showed a further reduction to 146 MPa. Furthermore, the tensile strength could be reduced by the addition of NR to the blend. The highest elongation at break was obtained for the NP1 80 blend with 403.7%.

On the other hand, the PHi002 blends exhibited a dramatical drop in E-modulus with the first addition of NR (20%) which resulted in similar mechanical properties to those observed for the original PHi001 compound, in spite of the lower percentage of modifier by missing PBAT in the PHA. As a consequence, a higher PHBV content led to a higher crystallinity, but to smaller, better-distributed crystalline domains, as can be seen from the SEM pictures in Section 3.6.

### 3.5. Thermal Properties

#### 3.5.1. DSC

The thermal properties were investigated for pristine NR and PHBV, as well as the NR/PHBV blends with various compositions.

According to the DSC results, NR had a glass transition around −46.6 °C. PHi001 and PHi002 had similar glass transition points at around −3.3 and 5.2 °C, respectively. This was well observed for PHi001 because of the very low crystallinity, whereas, for the Phi002 sample, more or less no *T_g_* was observed due to the high crystallinity. After the addition of NR, no significant *T_g_* change was observed in the various NP1 blends, whereas the *T_g_* decreased slightly with an increase in NR for the NP2 blends. The *T_g_* of NR was not determined in this investigation.

The degree of crystallinity was calculated according to Equation (1). Pristine PHBV (PHi002) gave 14% and 72% of X values, while, for NR/PHBV blends, the crystallinity fell to values in the range of 0.5–28%. This trend was due to the amorphous nature of NR, leading to a reduction in the crystallinity of PHBV [34]. Blends with a higher PHBV content also had a higher crystallinity. NR/PBHV blends with a ratio 80:20 displayed the lowest X values of 0.5% and 2%. NR/PHBV blends with ratios of 50:50 and 20:80 showed increasing crystallinities of 4%/19% and 7%/28%, respectively. These results underline the differences between the two PHBV samples PHi001 and PHi002, as well as the influence of the modifier contained in PHi001.

#### 3.5.2. TGA

Thermogravimetric studies can help to establish the conditions for the processing of thermoplastic materials. This is especially important for PHBV, which has a very narrow processing window. Figure 5 shows the TGA curves of the different materials. For PHi001, a two-step decomposition behavior can be observed (Figure 5a), corresponding to the two different components present in this material. The gradual weight loss increased with temperature starting at around 230 °C, and a complete degradation could be observed at around 420 °C, leaving a weight residue of 4.47%. A thermal instability of PHBV above 250 °C has been reported in previous studies [35,36]. The thermal degradation process involves chain scission, leading to a reduction in molecular weight. The processing temperature at 160 °C for the bio-blends used in this work was about 70 °C below the initial degradation temperature of the neat polymer. Dicumyl peroxide vulcanized NR started degradation at 290 °C, below which it was quite stable. At 476 °C a 100% weight loss was observed. PHi002 (Figure 6b) weight loss occurred in a one-step degradation process from 270 to 305 °C.

All blends showed a degradation in two stages, reflecting the different thermal stability of NR and PHBV. The addition of natural rubber to PHBV only slightly changed the thermal stability of the biopolymer blends. It can be observed that the thermal decomposition was shifted slightly to higher temperatures. The first thermal degradation, between 270 and 310 °C, was associated with PHBV degradation, while the next step was ascribed to the PBAT, as seen in the graph of PHi001. From this, the content of PBAT could also be estimated. Comparing PHi001 and PHi002, the influence of the PBAT additive in PHi001 could be observed as the second step of the Phi001 thermogram and a slight broadening of both degradation ranges. PHi002 shows the extremely narrow degradation temperature range typical for crystalline polymers such as PHB with comparatively low molecular weight. The second step, between 310 and 460 °C, was ascribed to PBAT, overlapping with the degradation range of NR [14,15]; therefore, it could not be observed separately.

### 3.6. SEM

Mechanical properties of blends depend strongly on the adhesion between the phases and their distribution within the blend. Thus, SEM analyses were conducted to explore the morphology of the blended materials at different magnifications. The surfaces of the NR/PHBV blends with a 50:50 ratio were very smooth with a phase-separated morphology, where PHBV was dispersed in the NR matrix (Figure 6). Additionally, SEM images were used to compare the blend surfaces before and after enzymatic treatment. The enzymatic hydrolysis of PHBV mainly proceeds via a surface erosion mechanism [37]. After the degradation experiments, the surface of the films showed morphological changes. The surface texture became much rougher for blends treated with depolymerase. In addition, pore formation was observed. Cracks, holes, and cavities were formed on the surface of the blends (b,d) indicating that *Pl*DP attacked the surface. In all SEM pictures, a stretching of the phase domains can be observed. This probably resulted from the processing of the compounds on the roll mixer. In Figure 6c,d, the finer graining of the PHBV phase because of PBAT’s absence can be better observed after degradation, which caused an effect similar to etching.

### 3.7. XRD Characterizations

In order to determine the crystalline nature of the NR/PHBV blends, X-ray diffraction analyses were performed. X-ray patterns of the pristine materials and crosslinked blends are depicted in Figure 7. NR showed the typical behavior of an amorphous polymer and displayed a broad hump in the 2*θ* range of 10.0° to 30.0°. Other studies observed similar results [38,39]. On the other hand, PHBV was highly crystalline and the characteristic diffraction peaks of PHBV could be observed at 13.3°, 16.7°, 19.8°, 21.4°, 22.4°, 25.3°, and 26.9° 2*θ*.

In general, the degree of crystallinity of the blends increased when PHBV was the major phase and decreased when the amount of NR in the blends increased (Figure 7a,b), confirming the DSC observations. Blends with 50% and 80% NR showed the broad peak of NR, which was superimposed by the typical diffraction pattern of PHBV. In the sample containing only 20% NR, the contribution stemming from NR in the diffraction pattern was only minor. In PHi001, the broad pattern of the amorphous PBAT partition could be observed, similar to that of NR. However, all the characteristic peaks observed in NR and PHBV were present in all blends, proving the immiscibility of these two polymers in all ratios. Moreover, it should also be noted that the peaks of PHBV were slightly narrower in the blends compared to the pristine sample. This indicated an increased crystallinity of the PHBV phase in the blends, induced by the processing at elevated temperatures (160 °C).

### 3.8. Toxicity Test

In this assay, MRC-5 cells were treated with different concentrations of eluates of the extracted samples, and the dehydrogenase activity was measured according to ISO 10993-5:2009 guide line. The results revealed no severe reduction in dehydrogenase activity, not even in the presence of the pure eluates (Figure 8). According to the definition of the ISO 10993-5 guideline, a decrease in viability below 70% of the untreated samples is defined as cytotoxicity. None of our samples caused cytotoxicity, not even the blends vulcanized with 6% crosslinker. The maximal decrease in viability was seen for sample NP2 50 4% with 89% ± 0.4%. For the blends without crosslinker, an increase in the dehydrogenase activity of the MRC-5 cells was observed. This could possibly indicate an improvement in viability, which needs to be further investigated.

### 3.9. Degradation

In nature, biodegradation of plastics is divided into three stages: biodeterioration, bio-fragmentation, and assimilation. Biodeterioration is also called surface-level degradation and represents the initial enzymatic attack of the smooth, hydrophobic plastic surface. This step is mostly achieved by the attachment of microbial communities and the formation of biofilms, which produce enzymes that are capably of cleaving molecular bonds between plastic molecules. This step already changes the mechanical, physical, and chemical properties of the material. Furthermore, it was shown that, once this step is fulfilled, biodegradation happens much more quickly, mostly due to the increased surface area and accessibility for enzymes [40].

A property of PHB materials is enzymatic degradation. Therefore, selected NR/PHBV blends were treated with the PHB depolymerase (*Pl*DP), and the degradation process was determined photometrically. Blends with 4% crosslinker were used. In general, the progress of degradation was faster for blends with 50% and 80% PHBV than for blends with only 20% PHBV (Figure 9). While degradation of NP1 with 50% and 80% PHBV (Figure 9a) and NP2 with 50% and 80% PHBV (Figure 9b) could already be measured after 4 h, blends with only 20% PHBV showed almost no degradation after 168 h. This can also be seen in Figure 9c, where the release of 3-hydroxybutyric acid during degradation led to an ionic shift of fluorescein, and samples become nonfluorescent. This retardation of degradation rates is a result of the high specificity of *PI*DP, which is only active for linear polyesters. Therefore, degradation rates were generally higher for NP1 blends, which was due to the PBAT content of the PHi001 material [41,42].

The degradation of NR needs different microorganisms or enzymes. Even though many microorganisms involved in latex degradation have been identified [43] and some enzymes have been characterized [44,45], the knowledge of the mechanistic basis of rubber biodegradation is highly limited. This is mostly due to the fact that these enzymes cannot be produced recombinantly, as presented in this study for PHA depolymerases, which is a prerequisite for such studies. Therefore, recombinant production of rubber-degrading enzymes followed by a more thorough characterization of the biodegradation of natural rubber or NR–PHB blends could be the focus of a follow-up study.

## 4. Conclusions

The purpose of this work was to generate and characterize a fully biobased and biodegradable blend with good mechanical properties and processability, based on two immiscible polymers: natural rubber and PHB blends. Our results prove that these bio-composites exhibit interesting properties such as high elasticity, as well as strength and durability. NR addition to a PHBV matrix had a positive effect on toughness and impact strength of PHBV, while PHBV addition to an NR matrix enhanced stiffness and impact resistance.

We also examined the biodegradation of NR–PHA blends enzymatically at lab scale by expressing selected depolymerase genes in *E. coli* and testing their activity after purification. As expected, depolymerases efficiently degraded the PHB partition of the blend, which we monitored on three different levels: first, by using a fluorescence-based activity assay; second, by investigating the surface properties of the samples; third, by examining the degradation products. The impact on surface smoothness was especially intriguing. After the addition of enzymes, we observed the appearance of cracks, holes, and cavities in the PHA fraction of the material. This may be an important first step for better and faster biodegradation of blends containing natural rubber, since rubber degradation is mostly limited by the fact that vulcanized materials are less accessible for the enzymes required for biodegradation. In general, efficient biodegradability adds great value to biobased materials in terms of application fields and eco-friendliness. Considering the superior properties of the PHB-NR blends described in this study, these materials may be ideal candidates for soft toys, such as small ducks, since outdoor toys are more prone to being lost coincidentally.

## Figures and Tables

**Figure 1 polymers-15-02593-f001:**
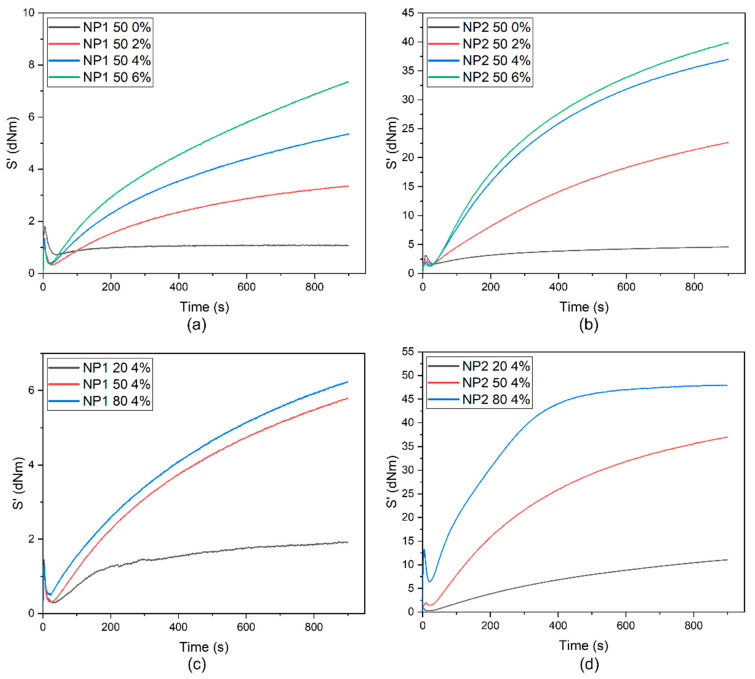
Rheometer cure curves at 160 °C of (**a**) NP1 50 blend with a 50:50 ratio and (**b**) NP2 50 blend with a 50:50 ratio. Different amounts of peroxide (Perkadox) were added (0–6%). (**c**) Rheometer cure curves at 160 °C of PHi001 blends with different amounts of NR crosslinked with 4% peroxide. (**d**) Rheometer cure curves at 160 °C of PHi002 blends with different amounts of NR crosslinked with 4% peroxide.

**Figure 2 polymers-15-02593-f002:**
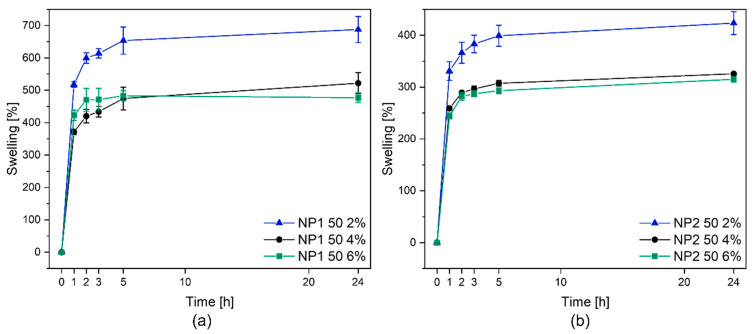
Swelling tests: crosslinked NR/PHBV (50/50) blends with different amounts of Perkadox were put into CHCl_3_ for 24 h at RT. Samples were weighed at specific times: (**a**) NR/PHi001 blends; (**b**) NR/PHi002 blends.

**Figure 3 polymers-15-02593-f003:**
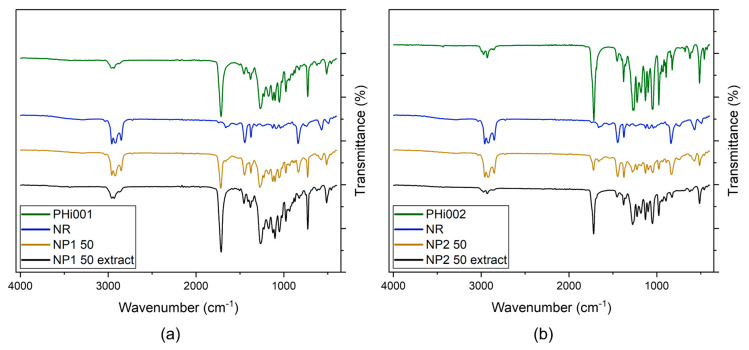
FTIR spectra of natural rubber (NR) and (**a**) PHBV (PHi001), vulcanized NR/PHBV (50:50) blend (NP1 50), and the extract of NP1 50 within the swelling experiment, or (**b**) PHBV (PHi002), vulcanized NR/PHBV (50:50) blend (NP2 50), and the extract of NP2 50 within the swelling experiment.

**Figure 4 polymers-15-02593-f004:**
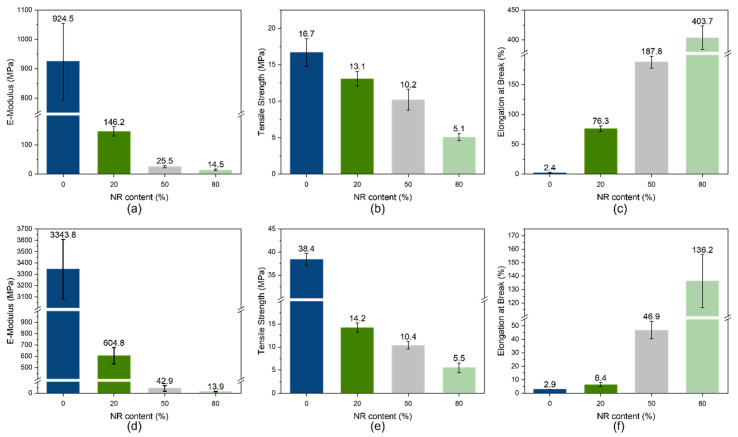
Mechanical properties of crosslinked NR/PHBV blends: (**a**–**c**) NR/PHi001 blends; (**d**–**f**) NR/PHi002 blends.

**Figure 5 polymers-15-02593-f005:**
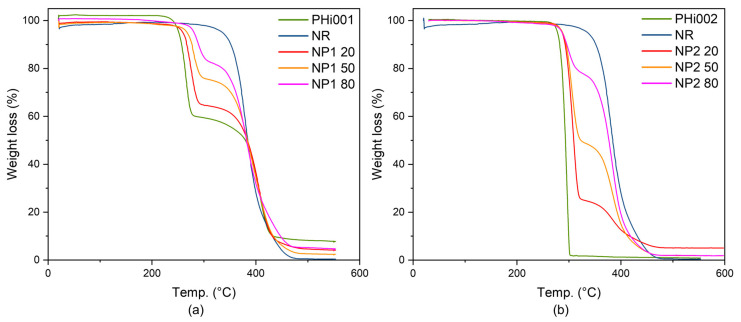
TGA thermograms of natural rubber (NR), PHBV samples, and various NR blends: (**a**) Compounds with PHi001 as base polymer, (**b**) PHi002 as base polymer.

**Figure 6 polymers-15-02593-f006:**
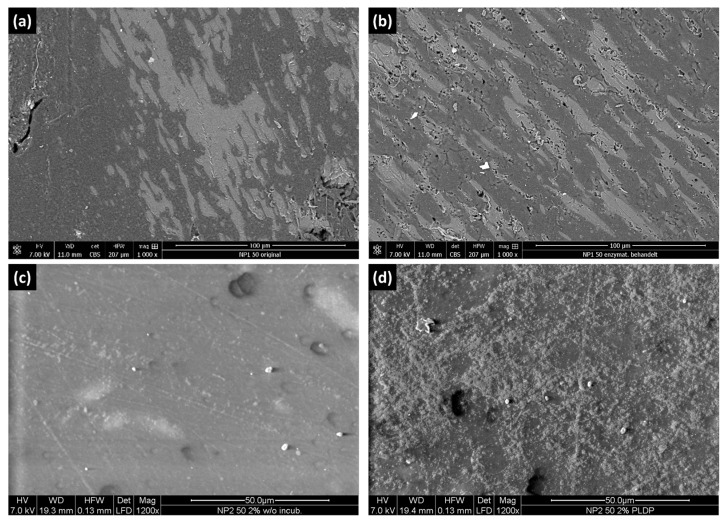
SEM micrographs of (**a**) NP1 50 blend with magnification of 1000× before degradation, (**b**) NP1 50 blend after degradation, (**c**) NP2 50 blend with magnification of 1200× before degradation, and (**d**) NP2 50 blend after degradation.

**Figure 7 polymers-15-02593-f007:**
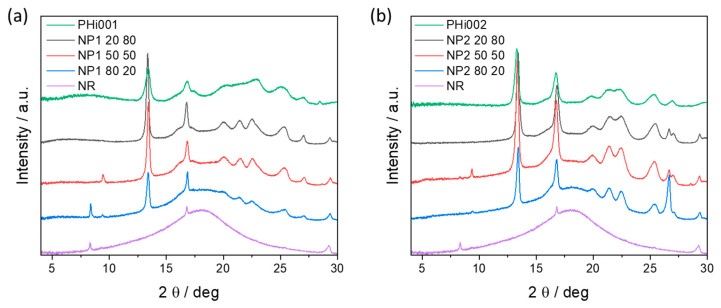
XRD patterns of (**a**) pristine PHi001, NR, and their various NP1 blends, and (**b**) PHi002, NR and their various NP2 blends.

**Figure 8 polymers-15-02593-f008:**
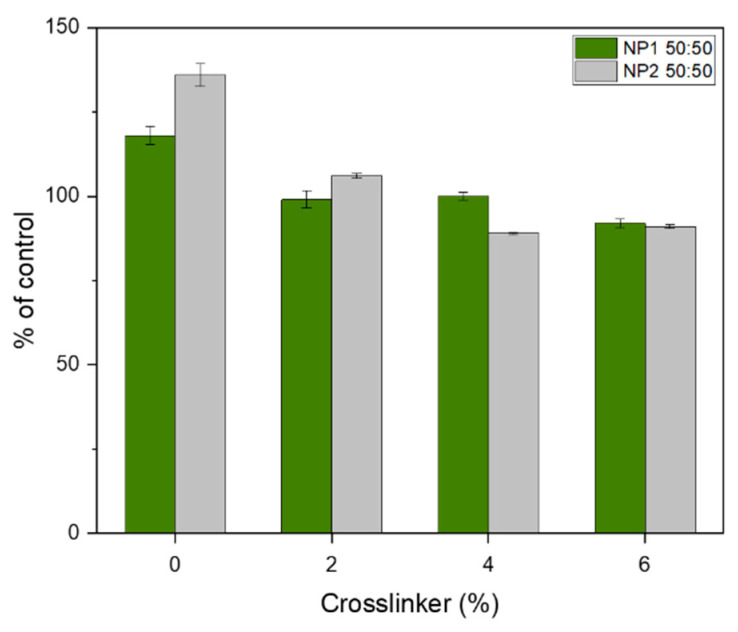
Dehydrogenase activity of MRC-5 cells exposed to NP1 and NP2 for 24 h. The viability of cells treated with medium was set to 100%. NP1 50:50 and NP2 50:50 with different amounts of crosslinker (0%, 2%, 4%, and 6%) were tested.

**Figure 9 polymers-15-02593-f009:**
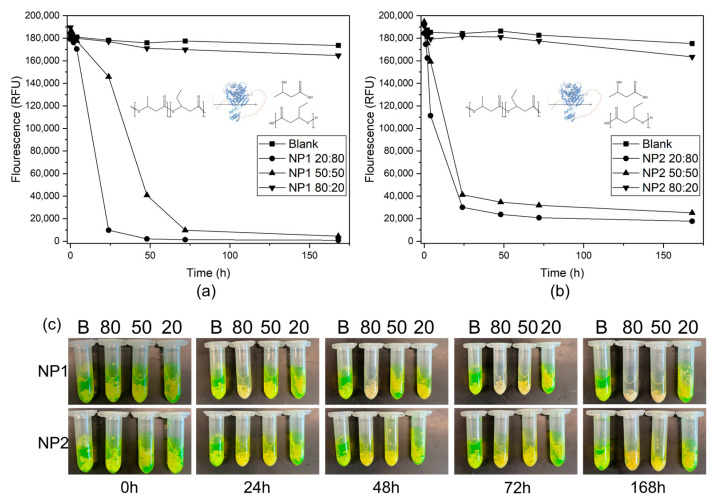
Enzymatic degradation of NR/PHBV blends by PHB depolymerase from *Pseudomonas lemoignei* (*Pl*DP) at 30 °C. (**a**,**b**) Fluorescence decreased during degradation via the release of 3-hydroxybutyric acid from NP1 blends (**a**) and NP2 blends (**b**). (**c**) Representative images of NP1 and NP2 blends treated for up to 168 h with 0.1 mg/mL.

**Table 1 polymers-15-02593-t001:** Composition of NR/PHBV blends.

Blend	Natural Rubber [%]	PHi001 [%]	PHi002 [%]
NP1 20	20	80	-
NP1 50	50	50	-
NP1 80	80	20	-
NP2 20	20	-	80
NP2 50	50	-	50
NP2 80	80	-	20

**Table 2 polymers-15-02593-t002:** DSC data of PHBV, NR, and the various blends.

Material	Δ*H_m_* [J/g]	*T_g_* [°C]	*T_m_* [°C]	*T_c_* [°C]	*X* [%]
PHi001	21.2	−3.3	160	50.8	15
PHi002	92.8	5.2	168	106.2	64
NR		−46.6			
NP1 20	10.6	−2.2	156.1	59.5	7
NP1 50	5.7	−3.6	155.1	57.7	4
NP1 80	0.8	−3.7	158.2	54.5	0.5
NP2 20	41.5	2.5	160.5	98.8	28
NP2 50	27.4	−1.9	161.4	94.3	19
NP2 80	2.7	−2.5	162	-	2

## Data Availability

The authors confirm that the data supporting the findings of this study are either available within the article or can be obtained upon request.

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
