# Peer review of "Bio-Polyester/Rubber Compounds: Fabrication, Characterization, and Biodegradation"

_polymers, 2023, doi:10.3390/polym15122593_

Round 1

Reviewer 1 Report

The article Bio-Polyester/Rubber Compounds: Fabrication, Characterization, and Biodegradation by Frank and coworkers reported the generation of a biodegradable polyester/natural rubber material with enhanced mechanical properties than PHAs and the characterizations of different properties of this material. Overall the article is well written and the data presented are of high quality.

I only have a couple of questions/suggestion for the authors.

1.    Page 7, line 276-278. The authors are claiming that the extractable part of the material is less than 1%. Is this quantification based on IR? Based on weight loss? It is not clear to me, and I don’t think IR is quantitative.

2.    Same question on page 8, line 311-314. Please clarify.

3.    Page 10, line 363-367, repetitive. The equation is already mentioned in the method section.  

Author Response

The authors want to thank Reviewer 2 for his valuable comments, we tried to follow all the recommendations as noted in red in the attached file.

Reviewer 2 Report

The article is devoted to an important problem: the creation of eco-friendly polymer compositions with outstanding properties. The authors created and studied mixtures of PHBV with natural rubber as a replacement for the traditionally used amorphous PHAs additive to reduce PHBV brittleness. The results obtained in the article will be of great interest to scientists working in the field of biodegradable polymers. I recommend this work for publication, but there are several comments. 

1. The authors need to clarify in section  2.1 if mixture PHi002 contains the same amount of PBAT and HV units as PHi001? Or is PHi002 without PBAT?

2. In section 2.12, it is necessary to clarify the conditions under which enzymatic degradation was carried out.

3. The distribution and size of PHBV phase domains affects dramatically the mechanical properties of the blends and their biodegradation. Note that the microstructures of PH1 50 and PH2 50 look different, even after enzymatic degradation (Figure 6). It is necessary to discuss this in the article.

4. The article should be carefully proofread, as there are some misprints, for example "whicht" (line 296), "incoluble" (313).

The article should be carefully proofread, as there are some misprints, for example "whicht" (line 296), "incoluble" (313).

Author Response

The authors want to thank Reviewer 3 for his valuable comments. We tried to answer all questions and follow the recommendations as noted in red in the attached file

Reviewer 3 Report

Overall, manuscript makes a good impression, and can be accepted after minor revision. I see only few minor remarks:

- Authors use attenuated total reflection (ATR) mode for FTIR analysis. Why standart absorption IR spectra in KBr was not used? 

-Authors have study the thermal decomposition and they observe a two-step mass lost. It's not obvious what the product of the termal decomposition? Both steps relate with chain scission or different derivatives of rubber are the place to be? I think products of thermal degradation should be specified (if it possible).

Author Response

The authors want to thank Reviewer 4 for his valuable comments. We tried to answer all questions and to follow all recommendations as noted in red in the attached file.

Reviewer 4 Report

The aim of this study was to create and evaluate a completely biobased and biodegradable blend, which has desirable mechanical properties and can be easily processed. This blend is made up of two immiscible polymers, namely natural rubber and poly(3-hydroxy-butyrate) blends. The findings demonstrate that these bio-composites display remarkable characteristics.

The subject is interesting and the manuscript has been well written. The findings have been effectively conveyed and thoughtfully analyzed. Furthermore, the methodology section has been meticulously outlined, ensuring replicability.

However, some revisions are required before final decision.

1. Please provide more references from Polymers.

2. The abstract requires significant revision to meet the necessary standards. In research papers, abstracts serve as a brief summary of the entire study. They should consist of the following components: i) a description of the topic's significance, current literature references, or identification of knowledge gaps; ii) the study's objectives; iii) a brief overview of the methods employed; iv) the main findings; and v) the implications of the results and the study's value. However, the current abstract format neglects the fourth and fifth components, which are critical.

3. Please do not use the acronyms in the keywords.

4. Please do not use the sub-headings in the Introduction and provide it in one section.

5. Inadequate literature survey characterizes the Introduction. Academic writing stands out for being grounded in pre-existing knowledge, previously conducted research, and established concepts and theories. It is imperative that the literature review is both methodical and analytical, scrutinizing and assessing the studies or ideas that pertain to the present work.

6. In order to ensure a comprehensive and informative introduction, it is recommended that the novelties and contributions of the study be presented in a distinct paragraph. It is crucial that this section addresses key questions such as the main research inquiries, any gaps in the field that the study addresses, and how it contributes to the subject area in comparison to existing literature. Without these details, it may not be clear what distinguishes the study from previous research.

7. What is the strategy behind the selection of these weight percentages of the materials (Table 1)?

8. Please provide the equation numbers; for instance, in line 159, 190, 203 and etc.

9. Please provide a supporting reference for Equation in line 159. The references from Polymers are in priority.

10. Please provide supporting references for Equations in lines 190 and 203. The references from Polymers are in priority.

11. Please provide the standard deviations for Figure 2.

12. Equation presented in line 159 has been repeated in line 364. Please resolve this issue.

13. Line 499 “shape-memory effects”, there are no results or discussion on the shape memory performance of the prepared samples in the text.

Author Response

The authors want to thank Reviewer 5 for his valuable comments. We tried to answer all questions and to follow all recommendations as noted in red in the attached file.

Reviewer 5 Report

The article is devoted to an important problem: the creation of eco-friendly polymer compositions with outstanding properties. The authors created and studied mixtures of PHBV with natural rubber as a replacement for the traditionally used amorphous PHAs additive to reduce PHBV brittleness. The results obtained in the article will be of great interest to scientists working in the field of biodegradable polymers. I recommend this work for publication, but there are several comments.

1. The authors need to clarify in section  2.1 if mixture PHi002 contains the same amount of PBAT and HV units as PHi001? Or is PHi002 without PBAT?
2. In section 2.12, it is necessary to clarify the conditions under which enzymatic degradation was carried out.
3. The distribution and size of PHBV phase domains affects dramatically the mechanical properties of the blends and their biodegradation. Note that the microstructures of PH1 50 and PH2 50 look different, even after enzymatic degradation (Figure 6). It is necessary to discuss this in the article.
4. The article should be carefully proofread, as there are some misprints, for example "whicht" (line 296), "incoluble" (313).

Author Response

The authors want to thank Reviewer 6 for his valuable comments. We tried to answer all questions and to follow all recommendations as noted in red in the attached file.

Round 2

Reviewer 4 Report

The revision is satisfactory. It is now acceptable.